# A Vibro-Acoustic Hybrid Implantable Microphone for Middle Ear Hearing Aids and Cochlear Implants

**DOI:** 10.3390/s19051117

**Published:** 2019-03-05

**Authors:** Ki Woong Seong, Ha Jun Mun, Dong Ho Shin, Jong Hoon Kim, Hideko Heidi Nakajima, Sunil Puria, Jin-Ho Cho

**Affiliations:** 1Department of Biomedical Engineering, Kyungpook National University Hospital, 130 Dongdeok-ro, Jung-gu, Daegu 41944, Korea; seongkw@ee.knu.ac.kr; 2Graduate School of Electronics Engineering, Kyungpook National University, 80 Daehak-ro, Buk-Gu, Daegu 702-701, Korea; mhj1990@naver.com; 3Institute of Biomedical Engineering Research, Kyungpook National University, 680, Gukchaebosang-ro, Jung-gu, Daegu 41944, Korea; swap9552@naver.com; 4Department of Medical & Biological Engineering, Graduate School, Kyungpook National University, 680, Gukchaebosang-ro, Jung-gu, Daegu 41944, Korea; jh850526@naver.com; 5Eaton Peabody Laboratory, Department of Otolaryngology-Head and Neck Surgery, Harvard Medical School, Massachusetts Eye and Ear Infirmary, 243 Charles Street, Boston, MA 02114, USA; Heidi_Nakajima@meei.harvard.edu (H.H.N.); Suni_Puria@meei.harvard.edu (S.P.)

**Keywords:** implantable microphone, hybrid vibro-acoustic sensor, electret condenser microphone, electret-type acceleration sensor, high sensitivity, wide frequency range

## Abstract

To develop totally implantable middle ear and cochlear implants, a miniature microphone that is surgically easy to implant and has a high sensitivity in a sufficient range of audio frequencies is needed. Of the various implantable acoustic sensors under development, only micro electro-mechanical system-type acoustic sensors, which attach to the umbo of the tympanic membrane, meet these requirements. We describe a new vibro-acoustic hybrid implantable microphone (VAHIM) that combines acceleration and sound pressure sensors. Each sensor can collect the vibration of the umbo and sound pressure of the middle ear cavity. The fabricated sensor was implanted into a human temporal bone and the noise level and sensitivity were measured. From the experimental results, it is shown that the proposed method is able to provide a wider-frequency band than conventional implantable acoustic sensors.

## 1. Introduction

Wearing a hearing aid is often associated with a certain degree of social stigma [1]. To overcome this, totally implanted hearing aids would be ideal [2]. However, currently available hearing aids, such as air-conduction hearing aids, cochlear implants, and semi-implantable middle ear hearing devices, are of the partially implantable types with an external microphone and sound processor and not free of stigma [2,3,4,5]. To date, various types of implantable microphones have been developed for totally implantable hearing devices with various degrees of surgical difficulty for implantation [5,6,7,8,9,10,11,12,13,14]. The implantable microphones used in the TIKI [8] and Carina [9] devices have the disadvantages of body noise (e.g., from chewing, facial vibrations) and sensitivity loss, especially high frequency due to the filtering effect of the skin. In contrast, the Esteem [9] has a piezoelectric microphone element that senses middle-ear motion, greatly reducing the influence of body motion noise. Additionally, the Esteem makes use of acoustic gain due to the ear auricle and the ear canal. However, its implantation requires a complicated surgical procedure and the sensor’s point of contact can become displaced [9].

Ko et al. [15,16] proposed a micro electro-mechanical system (MEMS) acoustic sensor, based on an MEMS acceleration sensor with capacitive electrodes, which attaches to the umbo and converts umbo displacement to an electrical signal. This device has the same advantages as the Esteem; e.g., motion noise reduction, prevention of acoustic attenuation loss due to skin thickness, and use of ear canal gain. Furthermore, coupling with the microphone and umbo is surgically easier because no bone drilling is needed. However, Ko’s acoustic sensor exhibits a low sensitivity below 0.8 kHz due to poor acceleration sensor characteristics [15]. Despite the lack of a good low frequency response, it is a potentially suitable microphone for cochlear implants. However, middle ear hearing devices require a good low-frequency sensitivity similar to that of acoustic hearing aids. In the future, accelerometer sensitivity may be improved, and this idea may become suitable. To date, an implantable acoustic sensor with broadband characteristics and improved sensitivity is not yet available.

In the human hearing pathway, sound pressure is increased by up to about 12 dB compared to the external environment by the pinna and the external auditory canal. The sound pressure of the middle ear cavity is attenuated by less than 3 dB after passing through the tympanic membrane [17]. The use of this residual sound pressure in the middle ear cavity improves the sensitivity of the acceleration-type microphone attached to the umbo.

In this paper, we propose a vibro-acoustic hybrid implantable microphone (VAHIM) with a broadband response by simultaneously sensing the sound pressure and mechanical vibration signal. Dummy mass of Ko’s capacitive acceleration-type sensor is replaced by a tiny electret condenser microphone (ECM) with a similar mass. This enables residual sound pressure inside the middle ear cavity to be combined with the vibration of the umbo for a wide-frequency response implantable sensor. The desired design frequency characteristics were optimized using finite element analysis (FEA) of a four-beam round-type elastic membrane. A flexible polyimide elastic beam printed circuit board (PCB) was fabricated and the frequency characteristics of the vibration beam and the hybrid acoustic sensor were verified. The sensitivity, dynamic range, and noise floor of the implantable microphone were measured in fresh human cadaver middle ears.

## 2. Middle Ear Cavity Implantable Acoustic Sensors

Surgically, the easiest implantable microphones, for cochlear or middle ear implants, are the single-point connection types that do not require the device to also be anchored to stable bone using hardware with a screw necessary in two-point connection microphone systems [2]. Ko et al. [15] developed an MEMS electrostatic capacitive acceleration sensor, which included an amplifying circuit, attached to the umbo directly behind the tympanic membrane (Figure 1a). As shown in Figure 1b, this sensor forms a variable capacitor between the two plane electrodes; i.e., the lower contacts the umbo and vibrates according to the acoustic vibration of the tympanic membrane and the upper contacts a dummy mass (M). To control the responses of the sensor, four silicon square beam springs formed by an MEMS process are located between the two electrodes. Because of the inertia of the dummy mass, which moves significantly less at frequencies higher than a certain cutoff, the gap between the two electrodes is modulated by the acoustic vibration of the umbo. A low-noise amplifier produces an output voltage from the capacitance change of the sensor.

In this acceleration acoustic sensor, the mass moves simultaneously with the umbo for input vibrations below a cutoff frequency. Therefore, there is almost no change in the gap between the upper and lower electrodes; this results in a low sensitivity at low frequencies. However, when the vibration frequency of the umbo increases to above the cutoff frequency, the vibration displacement is significantly reduced due to the inertia effect of the mass body. Therefore, the displacement of the gap between the lower (umbo side) electrode and the upper electrode (mass side), as well as the output amplitude, are increased.

## 3. Proposed Vibro-Acoustic Hybrid Implantable Microphone (VAHIM)

In VAHIM, the residual sound pressure inside the middle ear cavity is used to compensate for the low frequency degradation of the acceleration-type MEMS sensor, and to improve its overall sensitivity in the 0.1 to 10 kHz frequency range. Sound entering the ear canal is attenuated by less than 3 dB by the eardrum [15] and its vibration is not significantly changed by the loading of less than 25 mg at the umbo [18]. Therefore, in the middle ear, if an acceleration sensor with a sensitivity of −50 dBV and an additional acoustic sensor with a sensitivity of −40 dBV are placed, but the total weight is unchanged, a microphone sensitivity of at least −40 dBV can be achieved in a low frequency range. Therefore, we have developed a hybrid sensor with an acoustic sensor in place of a dummy mass to compensate for the high-pass response of the conventional capacitive acceleration MEMS-type implantable acoustic sensor. As shown in Figure 2a, the dummy mass (Figure 1b) is replaced by an ECM of a similar mass in the proposed sensor. An upper electrode is located beneath the substrate containing the vibration beam, and is separated by a gap distance (30 μm from the electret disc surface, which contacts the lower electrode position to the bottom plate. The output signal of the hybrid sensor is the sum of the acoustic signal from the ECM and the acceleration signal influenced by the mass of the ECM. However, in this case, this sensor provides two separate outputs without summing so that each sensor can be monitored individually.

### 3.1. Frequency Characteristics of the Capacitive Acceleration Sensor

The hybrid acoustic sensor combines the output of two sensors: the ECM, which has a flat frequency characteristic in the whole band; and the capacitive acceleration sensor, which has a low gain in a low frequency band, but a high gain in a high frequency band, with an upper cutoff frequency. When the hybrid acoustic sensor vibrates according to the motion of the umbo, the displacement of the ECM mass and the upper electrode mechanism connected to it is shown as a low-pass filter (LPF) characteristic *G_M_*(*f*) (Figure 3a). In this configuration, there is one resonance frequency peak at *f_ecm_*, the position and height of which are determined by the ECM mass, the stiffness of the vibration beam, and the energy loss factor. In this second order LPF system, by selecting the damping coefficient of the vibration beam of the hybrid sensor, resonance-peak overshoot can be minimized [19].

The major difference between the proposed sensor and previous designs shown in Figure 1b is that the dummy mass is replaced with the ECM mass. The vibration characteristics of the ECM mass coupled with the vibration beam can be described as a second order LPF that has a spring with elastic modulus k, mass M, and loss coefficient r. When a vibrator drives the hybrid sensor with constant sinusoidal displacement at the umbo and if the vibrator frequency is lower than the LPF cutoff frequency *f*_1_, the vibration amplitude of the ECM mass is almost the same or higher than that of the vibrator. At frequencies exceeding the *f_n_*, the displacement of the ECM mass is decreased to almost zero due to the increase in the inertia effect of the ECM mass with increasing frequency. Therefore, at frequencies higher than *f*_2_, which is low-cutoff frequency of the capacitive acceleration sensor *G_CAS_*(*f*), the displacement amplitude of the umbo vibration corresponds to the gap distance of the plane capacitor. Thus, the frequency response of capacitive acceleration sensor *G_CAS_*(*f*), which has a relative output response to the displacement change of the gap due to the vibration between frequencies from 0 Hz to the upper cutoff frequency *f*_2_ can be given as:(1)$$G_{CAS}=\left|1-G_{M}\left(f\right)\right|,$$
and this response is shown in Figure 3b. The absolute value is taken on the right side of this equation because the displacement of the upper electrode changes from the original position to both the upper and lower side at the time of resonance. The *f*_2_ of the acceleration sensor is formed by coupling of the umbo and the small mass m, which is the sum of the bottom plate of the hybrid sensor, the upper and lower connecting pins, and the lower electrode. This m is a much smaller mass than the ECM mass M, but the loading effect increases as the driving frequency increases. Therefore, as the frequency increases above the *f*_2_, the effect on the gap distance variation of the plane capacitance due to umbo displacement weakens rapidly.

Finally, the overall response of the hybrid vibro-acoustic sensor *G_HVA_*(*f*) is plotted in Figure 3c. *G_HVA_*(*f*) is the combination of the responses in Figure 3b: solid line characteristics and dotted line characteristics, which represent the ECM gain with a uniform response in the desired frequency domain (0.1~10 kHz). In this case, the maximum response of the ECM is assumed to be 30% greater than that of the capacitive acceleration sensor. In Figure 3b, the gain of *G_MC_*(*f*) is very small below the *f*_1_. In other words, the low frequency gain is very small except for the distortion caused by the resonance peak in this band. However, as shown in the graph of the total gain *G_MCE_*(*f*) in Figure 3c, the low frequency gain below *f*_1L_ shows more than 57% of the maximum gain between *f*_1L_ and *f*_2H_. This also demonstrates that the greater the ECM gain, the more the low-frequency gain of the *G_MCE_*(*f*) improved compared to *G_MC_*(*f*). In Figure 3b,c, the upper cutoff frequency is indicated, and the dotted ECM characteristic generally has a higher upper cutoff frequency than the acceleration sensor. The resonance peak of the upper electrode displacement *G_M_*(*f*) may include distortion of the gain characteristic in the vicinity of 1.5 kHz (Figure 3c). This distortion problem can be solved by designing the springs and attenuation factor to ensure critical damping.

In this hybrid sensor, the actual output of the capacitive acceleration sensor part depends on the amount of Δ*C*, which is the capacitance change between the two electrodes. Additionally, Δ*C* is proportional to *y*, the vibrational gap displacement of the umbo:(2)$$\Delta C=\frac{\varepsilon S}{Y_{0}^{2}}y,$$
where *ε* is the dielectric constant of the inner medium of the plain capacitor, *S* is the opposing area of the electrode plane, and *Y*_0_ is the gap interval [10].

To convert Δ*C* into a signal voltage, we connected the upper electrode to the gate of the low noise field effect transistor (FET) integrated circuit (IC) of the capacitive acceleration sensor preamplifier.

### 3.2. Vibration Beam Plate Design

The low cut-off frequency *f*_1_ of the gain characteristic curve of the acceleration sensor (Figure 3b) depends on where the resonance frequency *f_n_* of the vibration beam is set. In this paper, we arbitrarily set *f_n_* at 1.5 kHz only for the purpose of confirming the effect of the ECM instead of the dummy mass of the acceleration sensor. In general, the vibration beam is determined by parameters such as the thickness, width, length, and number of the beams shown in Figure 4. By varying the parameters of the vibration beam, resonance points can be generated in the desired frequency band. The stiffness *k* and resonant frequency *ω_n_* of the vibrating beam are given by Equations (3) and (4), respectively [20].
(3)$$k=n\left(\frac{3EI}{l^{3}}\right)=\frac{nEWT^{3}}{4l^{3}},$$
(4)$$w_{n}=\sqrt{\frac{3EI}{ml^{3}},}$$
where *k* is the stiffness coefficient of the vibrating beam, *n* is the number of beams, *E* is the modulus of elasticity of the beam, *I* is the moment of inertia of the cross-section of the beam, *W* is the width of the beam, *m* is the mass of the ECM, and *T* is the thickness of the beam.

A circular four-beam vibration plate that responds to acceleration and is of a sufficiently low mass is shown in Figure 4. For this reason, materials lighter than metals and more flexible than silicone are advantageous. Thus, we designed a four-beam vibration plate of flexible thin polyimide, which acts as a vibrating beam with a mounted PCB with a low noise amplifier IC chip for a capacitive acceleration microphone.

To design the frequency characteristics of the beam plate located immediately below the ECM, the mass applied to the beam plate, the beam structure, and the elasticity need to be considered simultaneously. Therefore, the characteristics of the vibrating beam were analyzed using FEA software (COMSOL Multiphysics 5.0; COMSOL Inc., Stockholm, Sweden) to obtain values more precisely than the first order approximations yielded by the simpler formula. For the FEA, the structure of the hybrid vibro-acoustic sensor was analyzed by dividing it into meshes (Figure 5a), and subjected to three-dimensional vibration displacement analysis (Figure 5b). The mesh of the sensor model consisted of 233,105 domain elements, 22,532 boundary elements, and 1992 edge elements using a defined “free tetrahedral”.

To perform the analysis, the angle *θ* of the beam was changed from 30° to 60°. The width *W* of the beam was fixed to 350 µm and the beam thickness *T* was set to 200 µm for the planned PCB production. As shown in Figure 5c, resonance occurs at 1.5 kHz when *W* = 350 µm, T = 200 µm, and angle = 50°. As the length of the beam decreases, attaching the connecting pins to the beam plate becomes more problematic. Therefore, the structure of the vibration beam was set to a 350 µm width, 200 µm thickness, and 50° angle based on the error range of the manufacturing process. To provide a signal input/output function on one side of the four-beam elastic bodies, 10 µm thick gold lines are printed. However, because these lines are markedly thinner than the polyimide, their effect on the migration of the resonance peak is insignificant.

### 3.3. Fabrication of the Hybrid Vibro-Acoustic Sensor

Based on the results of the FEA, the vibration beam plate printed circuit board (PCB) was fabricated using polyimide material (Figure 6a,b; blueprint and actual product, respectively). The PCB had a 3 mm diameter and four round vibration beams were present on the edge of the vibration plate. Bonding electrodes are disposed at the center of the vibration plate so that a low-noise amplifier IC chip (LMV1032UR-25, Texas Instruments, Dallas, TX, USA) for signal amplification can be fixed in an area with a 2 mm inner diameter. This hybrid sensor uses the ECM (OBG-311L42-C33, BSE Co., Ltd., HongKong) as a mass at the center of the vibration plate to respond to the acceleration of the umbo vibration. Furthermore, a plane capacitor is formed between the circular electrode on the underside of the vibration plate and the circular electrode on top of the bottom plate. The ECM of the uppermost layer and the vibration beam plate beneath it are electrically connected so that the output signal arrives at the bottom plate. The printed circuit board (PCB) patterns are designed to connect the signal and power lines of the ECM, and the top and bottom of the vibration plate and the bottom plate.

The vibration and bottom plates are electrically connected via four conductive miniature connecting pins standing vertically on the bottom plate. The four output terminals on the bottom plate are configured to pull out one ECM output signal, one capacitive acceleration sensor output signal, two power lines for driving the ECM, and the integrated circuit (IC) for the capacitive acceleration sensor.

The VAHIM components and the final assembly are shown in Figure 7. The VAHIM is cylindrical, with an outer diameter of 3.1 mm and height of 2.8 mm. Immediately below the ECM is a 0.1 mm thick plastic insulation plate, and the power and signal lines of the ECM are connected through the holes in the insulation plate to the PCB pattern above the vibration beam plate. The bottom surface of the vibrating beam substrate is provided with an IC chip as a preamp and a plate electrode for detecting any change in capacitance. An electret disc is attached to the upper surface of the bottom plate to increase the sensitivity of the acceleration sensor according to the capacitance change, without using an external battery. The four vibrating beams at the edges of the vibrating beam plate are connected to the four connecting pins. Since the gap between the electrodes in the acceleration sensor is controlled by the length of the connecting pins, some laser trim processing is required.

## 4. Experimental Results and Discussion

### 4.1. Bench Test of the Proposed Acoustic Sensor

First, the resonance frequency characteristics of the isolated vibration beam itself were measured. A piezoelectric element was attached to a fixed plate coupled to the bottom center of the vibration beam plate, and a mass of the same weight as the ECM was attached to the top center. Next, a constant external displacement was applied using a piezoelectric device, and the vibration displacement of the vibration beam was measured using a laser Doppler vibrometer (LDV). The signals were controlled and measured using PCI Extensions for Instrumentation (PXI) and LabVIEW™ (National Instruments Co., Austin, Texas, USA) software. The displacement of the vibration beam was measured at a constant vibration amplitude of 100 nm in the entire audible frequency range driven with a piezoelectric element. The frequency characteristics of the measured vibration beam driven by constant displacement at all audible frequencies and the vibration characteristics of the piezoelectric element are shown in Figure 8. The piezoelectric element applied an almost constant external force of 600 nm and the very small resonance point was 1.4 kHz. The measurement result with the fabricated vibration beam plate was similar to model simulation in Figure 5b, but with an unexpected dip of less than 3 dB in the 3 kHz region, indicating fabrication process errors. However, this was considered to be minor in comparison to the overall frequency response.

In general, the microphone sensitivity is reported as the voltage response of the microphone for an input level of 94 dB SPL (sound pressure level) as a function of frequency. Similarly, the sensitivity of the acceleration sensor is expressed by the output voltage of the sensor for a constant vibration input.

Because of the mixed units, it is difficult to directly compare the sensitivity of these two sensors. A direct comparison of these sensitivities is facilitated when the hybrid sensor is installed directly in-situ at the umbo. Thus, the comparison of the output characteristics of the two sensors for the same acoustic input is presented in the cadaveric measurement results section of this paper. Figure 9 shows the frequency response of the hybrid implantable acoustic sensor in the open space. The black solid line in Figure 9a shows a flat frequency characteristic of the ECM with a −40 dB sensitivity measured by the application of a sound of 94 dB SPL in the open space. The black dash line in Figure 9 shows the capacitive acceleration sensor measured by applying 300 nm displacement. The ECM frequency response was almost flat at 0.1 to 10 kHz, but the response of the capacitive acceleration sensor shows that the low-frequency gain was significantly low, at below 1.5 kHz, as predicted above. The acceleration sensor shows a sensitivity of −43 dB in the flat frequency band above the low cutoff frequency. Because the proposed sensor uses the combined output of these two sensors, the gain is increased. Furthermore, use of this hybrid implantable sensor would improve the bandwidth and gain compared to existing capacitive acceleration sensors used alone.

### 4.2. Human Cadaver Measurement Methods

The VAHIM was installed in the middle ear of a human cadaver and its characteristics were compared with those of a conventional acceleration sensor. We performed the measurements at the OtoBiomechanics Laboratory of the Massachusetts Eye & Ear Infirmary, Harvard Medical School. The experimental setup on a vibration isolation table is shown in Figure 10. SyncAv (ver 0.30) was used to generate stepped tones from 0.1 to 10 kHz applied to an NI PXI system, power amplifier, and the ear canal through a sound source. SyncAv simultaneously measured the voltage sensitivity of the implantable microphone and of the ER-7C microphones. The artificial ear canal was shielded with clay to prevent ingress of external sound. An ER-7C probe-tube microphone was placed within 3–5 mm of the eardrum and a second ER-7C measured the SPL in the middle ear cavity. For convenience, the incus was removed and the LDV was used to measure the displacement of the umbo. The VAHIM was mounted on the malleus manubrium close to the umbo in the middle ear cavity side of the cadaver with the experimental setup shown in Figure 11. Measurements were made on four temporal bones for this study. Some ears were used for practice placement and some had only partial results, while a complete set of measurements was successful in one ear and is reported here.

### 4.3. Human Cadaver Measurement Results

The VAHIM was attached to the malleus manubrium near the umbo. A plastic tube of a length matching that of the actual ear canal was inserted to form an artificial ear canal and covered on the lateral end with a glass cover slip for optical access to the umbo. A sound source was connected through a second smaller side tube. SyncAv was used to synchronously measure responses that included the sound pressure in the middle ear cavity, the vibration displacement of the umbo, and the output voltages of VAHIM. Approximately constant SPLs of 100, 94, and 88 dB were applied to the tympanic membrane using the StimEq functionality of SyncAv. The output response of the ECM and capacitive acceleration sensor are shown in Figure 12a,b respectively. The output of the ECM and the capacitive acceleration sensor are shown to be linearly independent of the applied sound pressure. The overall dynamic range of the ECM sensor was at least 82 dB at 1 kHz relative to the noise floor, and that of the acceleration sensor was 67 dB. The dynamic range of the acceleration sensor was about 15 dB lower than that of the ECM because of possible additional noise due to the lack of packaging and EMI shielding. The ECM and acceleration sensor output characteristics were compared following the application of 94 dB SPL to the eardrum.

We next compare the output voltage of the ECM and capacitive acceleration sensor independently and in combination for the same input sound pressure applied at the eardrum (Figure 13a). The output sensitivity of the ECM (blue line) in comparison to the acceleration sensor (red line) is attenuated by about 6 dB below about 1 kHz. The sensitivity output of the acceleration sensor (blue line) increases in comparison to the ECM above about 3 kHz. A possible reason for the difference between the open space response of the sensors and the cadaver insertion response is the additional effect of the cadaveric middle ear response to the hybrid sensor.

The output characteristics of the hybrid combined microphone are the sum of responses from the two sensors. Because the amplitude and phase-angle data of the two signals are stored simultaneously in SyncAv, they are combined to obtain the sum signal of two outputs and also shown in Figure 13 (black line). The sums of the signals were typically higher than that of either signal alone (Figure 13a, black solid line). However, for practical application, the final system will require a summing circuit, which can be simply achieved by using operational amplifiers.

The output of the combined sensor output and noise floor are shown in Figure 13b. Also shown for comparison are Ko’s sensitivity and noise results from a simulation model calculation (not in a cadaveric experiment). The measured noise level averaged across the frequency of the proposed sensor was about 5 dB lower than that of Ko et al., and the sensitivity of Ko et al. was 1.5 dB higher. The maximum and minimum of the signal-to-noise ratios were 78 dB (6000 Hz) and 44 dB (700 Hz) in Ko ’s study [15] and the proposed sensors were 79 dB (2000 Hz) and 50 dB (100 Hz), respectively. The average signal-to-noise ratios of Ko et al. and proposed sensor were 66 and 64 dB, respectively. However, the proposed sensor is very flat compared to Ko’s sensor in the entire audible band, which is a desirable feature for a hearing aid microphone.

When using this hybrid vibro-acoustic sensor as a hearing aid microphone, the noise caused by body motions, such as the shocks or fricatives that can flow through the skull and the sound of chewing in the middle ear cavity, should be considered. Body noise propagated through bone to the middle ear cavity and sensed by the ECM affects the acoustic component more so than the vibrational component propagating through the ossicles, which are isolated by ligaments and muscle. However, this may be overcome by adaptive signal processing using the acceleration sensor, which is less influenced by body motion noise, as a reference signal. In addition, errors may occur when the sensor is contained within a package; indeed, such an error is evident in the acceleration characteristics (Figure 12b). In other words, the sensitivity of the acceleration sensor should be capped, as in the open space test (Figure 9), to entire frequencies. This is because the sound pressure signal leaks into the capacitive acceleration sensor. Therefore, the electrode gap distance of the capacitive acceleration sensor is influenced by the sound pressure signal, particularly below 1.5 kHz in this case, so the capacitive acceleration sensor must be completely shielded to prevent the ingress of sound signals other than the vibration signal. This underlines the importance of isolation packaging between the ECM and the capacitive acceleration sensor. Another area of further improvement is to reduce the total weight of the microphone from 50 mg to less than 30 mg, which will require further iterations of the device design.

## 5. Conclusions

Here, we propose a broadband vibro-acoustic hybrid implantable microphone sensor for totally implantable hearing aids and cochlear implants. Our new sensor improves on the frequency characteristics, sensitivity, and noise floor of the old capacitive acceleration sensor [15,16] and is a single-point attachment to the umbo. This design both simplifies surgical implantation and enables the acoustic gain of the external ear canal and pinna to be utilized. To achieve this, we replaced the old capacitive acceleration sensor [15,16] with an ECM of an identical mass. Thus, a VAHIM with improved sensitivity and a wide-frequency band was realized by incorporating both an ECM with a flat frequency response and a capacitance acceleration sensor with a high sensitivity at high frequencies. The hybrid implantable acoustic sensor design was optimized by using finite-element modeling. The vibration beam, which had a resonance of 1.5 kHz, was fabricated from a polyimide material. We verified the usability of the implantable acoustic sensor by showing that the ECM sensor compensates for the limited low-frequency characteristics due to the mechanical coupling system of the mass and the vibration beam. The measured response of the VAHIM has an amplitude dynamic range of about 82 dB SPL at 1 kHz with improved sensitivity and the frequency dynamic range was extended in comparison to the conventional capacitive acceleration sensors. Further studies using modeling simulation and cadaveric experiments on the changes in the frequency characteristics of the sensitivity depending on the coupling strength between the sensor and the umbo are needed.

## Figures and Tables

**Figure 1 sensors-19-01117-f001:**
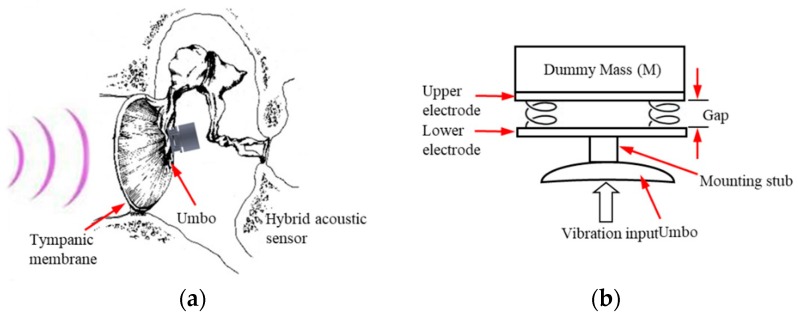
(**a**) Configuration of the micro electro-mechanical system (MEMS) electro-capacitive acceleration sensor for totally implantable hearing aids developed by Ko et al. [15]; (**b**) schematic for illustrating the principle of the MEMS capacitive acceleration acoustic sensor.

**Figure 2 sensors-19-01117-f002:**
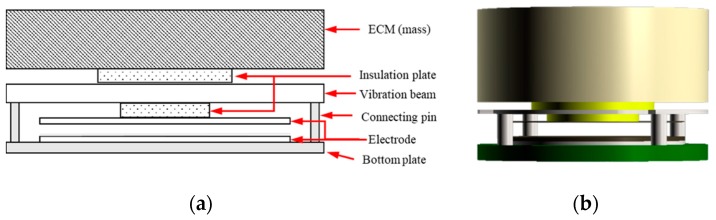
Vibro-acoustic hybrid implantable microphone (**a**) composition diagram and (**b**) assembled configuration. ECM: electret condenser microphone.

**Figure 3 sensors-19-01117-f003:**
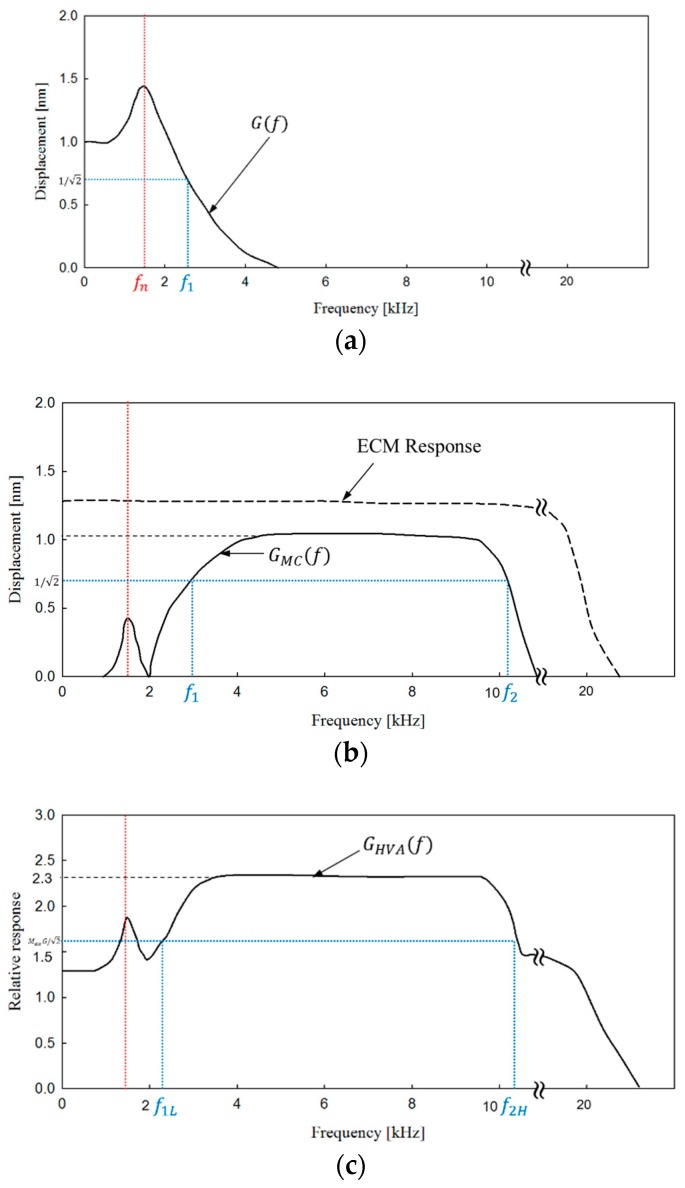
(**a**) Frequency response of the electret condenser microphone (ECM) mass connected with the upper electrode mechanism; (**b**) the solid line and the dashed line represent the response of the capacitive acceleration sensor and the ECM response, respectively; (**c**) overall response of the hybrid sensor, obtained by summing the responses of the capacitive acceleration sensor and the ECM.

**Figure 4 sensors-19-01117-f004:**
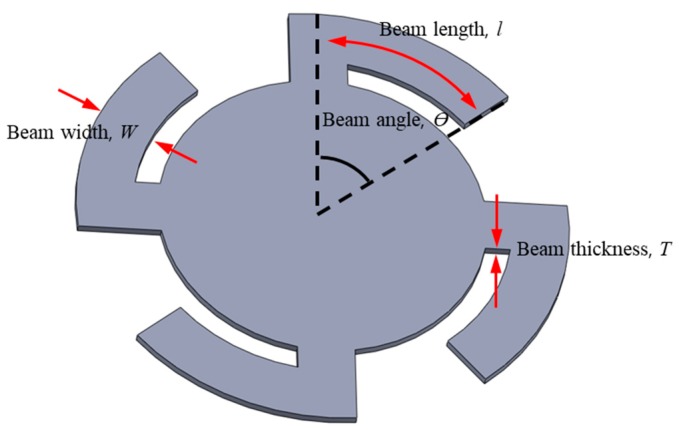
Vibration beam plate structure of the capacitive acceleration sensor.

**Figure 5 sensors-19-01117-f005:**
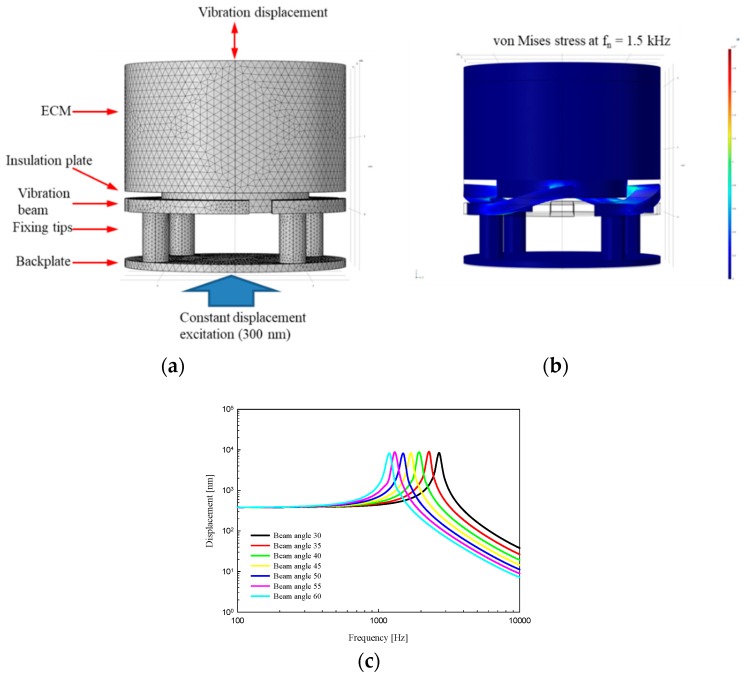
Finite element analysis. (**a**) Mesh distribution; (**b**) result of 3D vibrational analysis; (**c**) frequency characteristics with beam angle as a variable.

**Figure 6 sensors-19-01117-f006:**
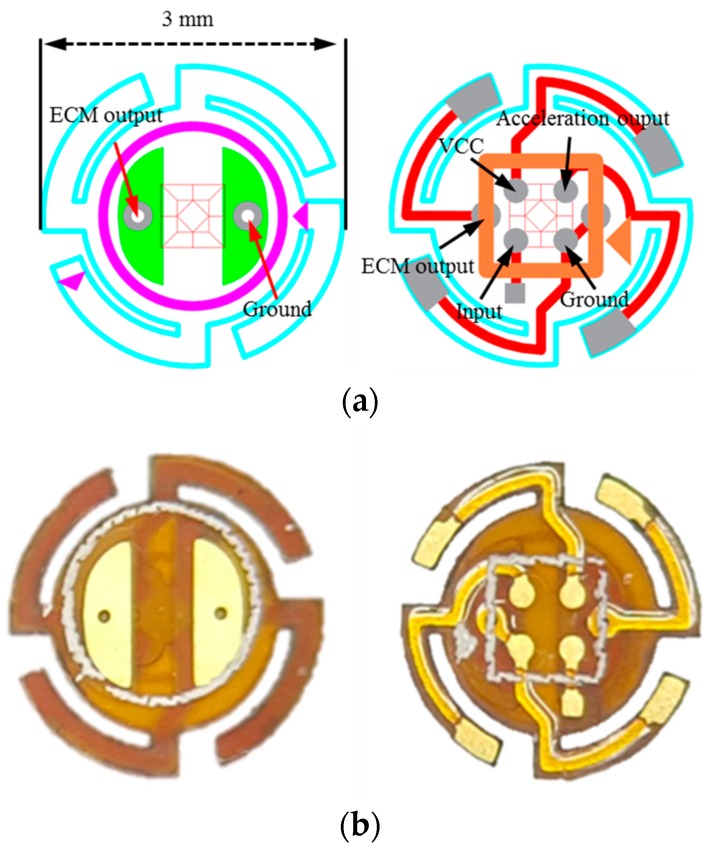
Flexible printed circuit board (PCB) of the vibration beam plate (**a**) blueprints and (**b**) manufactured PCB.

**Figure 7 sensors-19-01117-f007:**
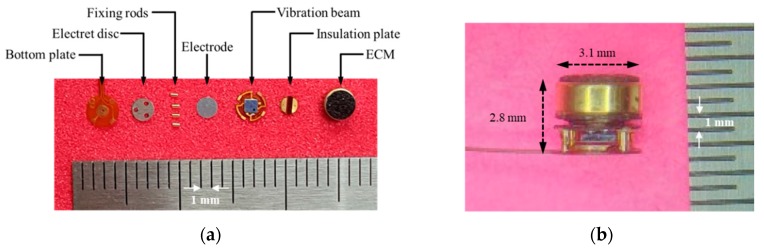
The vibro-acoustic hybrid implantable microphone (**a**) disassembled components and the (**b**) assembled part.

**Figure 8 sensors-19-01117-f008:**
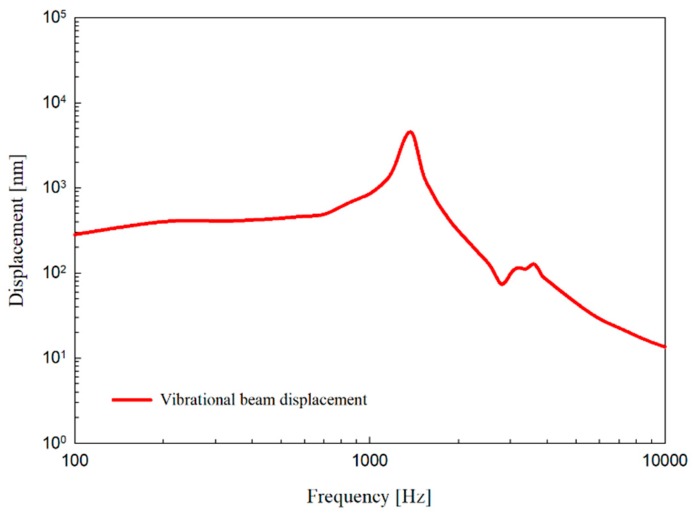
Vibration measurements of the the vibration beam attached to piezoelectric element.

**Figure 9 sensors-19-01117-f009:**
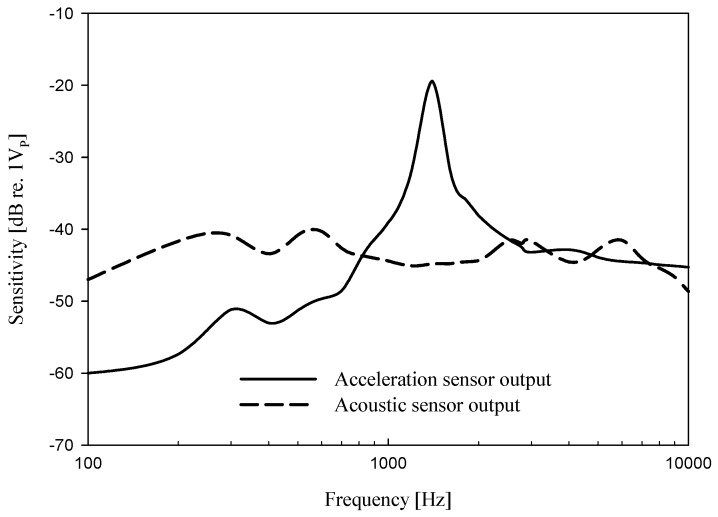
The voltage response in dB re 1 *V_peak_* of the microphone components measured independently.

**Figure 10 sensors-19-01117-f010:**
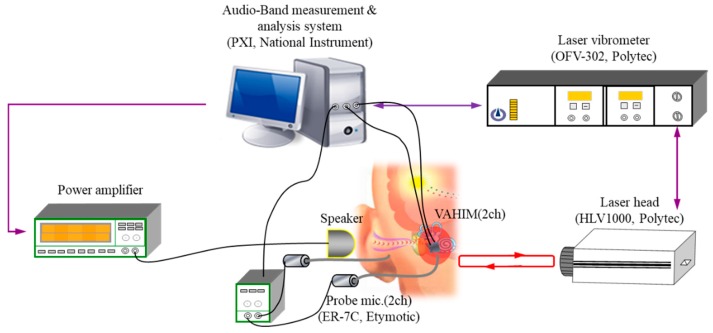
Measurement setup for cadaveric experiments. VAHIM: vibro-acoustic hybrid implantable microphone.

**Figure 11 sensors-19-01117-f011:**
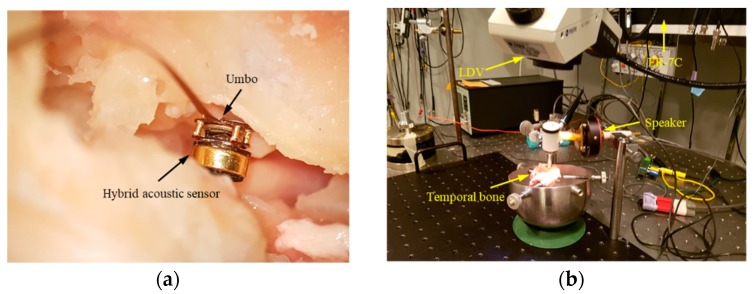
(**a**) Microphone attached at the umbo of a human cadaver temporal bone; (**b**) a human temporal bone experimental setup.

**Figure 12 sensors-19-01117-f012:**
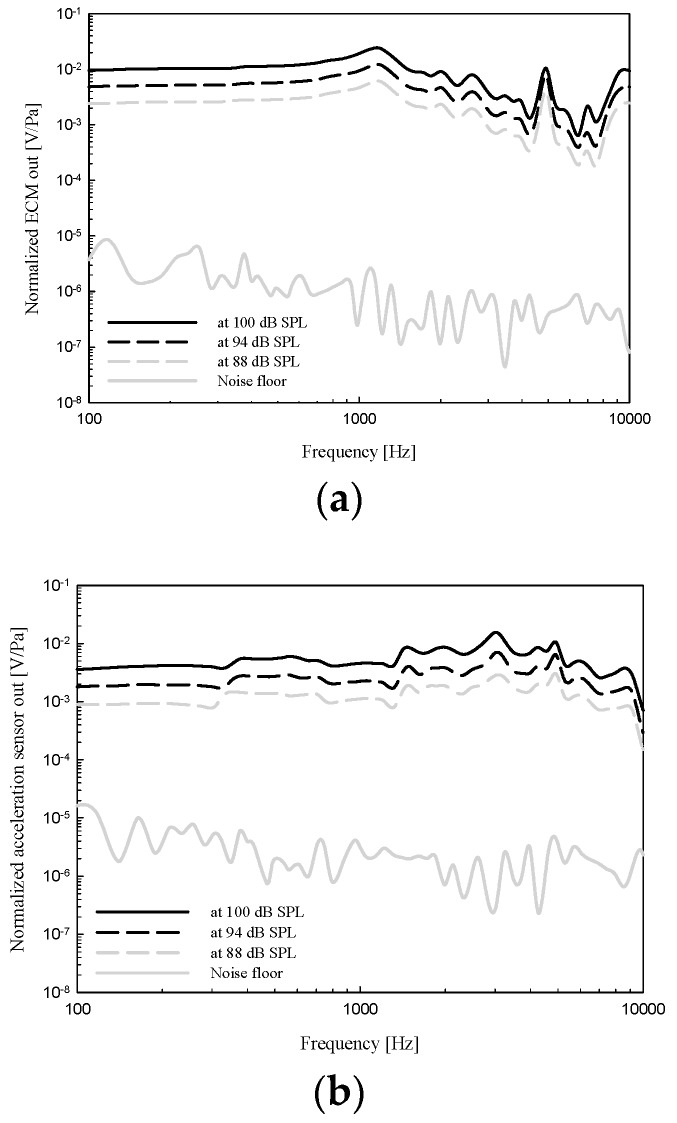
In-situ output response of the ECM (**a**) and capacitive acceleration sensor (**b**) measured 6 dB apart. Also shown are the corresponding noise floor measurements.

**Figure 13 sensors-19-01117-f013:**
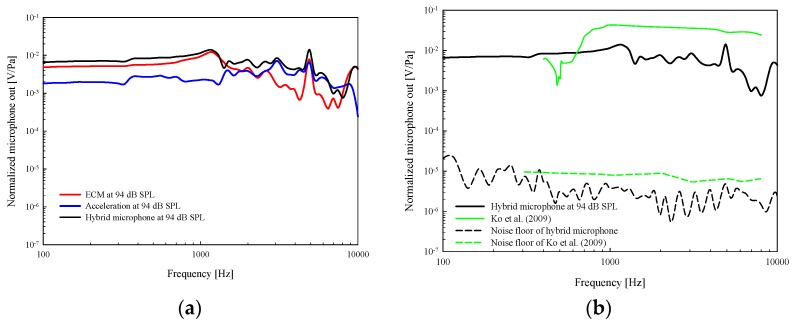
(**a**) Experimental sensitivity measured in a cadaveric temporal bone of the ECM (**red**), acceleration sensor (**blue**), and the combined responses of the two sensor components (**black**). (**b**) Comparison of the combined sensor response sensitivity and noise floor with previous study (Ko et al.).

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
