# Peer review of "A Vibro-Acoustic Hybrid Implantable Microphone for Middle Ear Hearing Aids and Cochlear Implants"

_sensors, 2019, doi:10.3390/s19051117_

Round 1
Reviewer 1 Report
In this manuscript, authors proposed a hybrid implantable acoustic sensor by combining electret condenser microphone (ECM) and capacitive acceleration sensor to improve performance. From the viewpoint of future acoustic sensing application, this manuscript seems interesting, and could be accepted by Sensors. However, several lacks of details can be found as follow:
1. The development of MEMS sensing device is an area that has received enormous attention. Therefore, the literature must be updated; more recent references must be incorporated.
2. In Introduction, authors only mentioned about different hearing aid devices and acoustic sensors with their disadvantages. Authors should also compare the their detailed performance with proposed hybrid sensor in results and discussion, such as dynamic frequency ranges and correlated sensitivities.
3. The schematic in Figure 2 is very busy. Authors should provide a clearer illustration to help readers to understand the sensor structure.
4. In numerical study, the mesh independence study is missing in the current manuscript.
5. Besides of sensitivity, information about the quantification limit, reproducibility, repeatability, stability, and their correlated error bars must be included.
Some minor typos are shown below:
(a). In line 217, it should be “the angle θ of beam was changed from 30 degree to 60 degree”, instead of “30 degree to 5 degree”.
(b). In line 219 and 220, it should be “W=350um, T=200um, and θ=50 degree” instead of “length = 50 degree”
(c). In line 222, authors should double check the length. Is it really 50um? From Figure 6 (B), it does not look like that.
Author Response
Reviewer #1
R1-1. The development of MEMS sensing device is an area that has received enormous attention. Therefore, the literature must be updated; more recent references must be incorporated.
Response to R1-1: We did another search for references to implantable acoustic sensors based on capacitive MEMS sensors implanted at umbo and other middle ear anatomy. For acoustic sensors with capacitive MEMS technology, only the reference by Wen Ko et al. (2009) turned up. This is now added to the reference list.
R1-2. In Introduction, authors only mentioned about different hearing aid devices and acoustic sensors with their disadvantages. Authors should also compare their detailed performance with proposed hybrid sensor in results and discussion, such as dynamic frequency ranges and correlated sensitivities.
Response to R1-2: Agreed. Conventional sensors have many problems such as body noise pick-up and surgical difficulties. Umbo attachment sensors (e.g., Ko et al.) provide solutions to these problems. In this paper, we have added a comparison Figure (13b) with Ko's study even though their results from a simulation model rather than in from biological tissue. The following text in Figure 13b was added to 3rd parapraph in P. 13.
<Revised> from P.13 Line 397, Figure 13. The output of the combined sensor output and noise floor are shown in Figure 13b. Also shown for comparison are Ko's sensitivity and noise results from a simulation model calculation (not in a cadaveric experiment). The measured noise level averaged across frequency of the proposed sensor was about 5 dB lower than that of Ko et al., and the sensitivity of Ko et al. was 1.5 dB higher. The maximum and minimum of the signal-to-noise ratios were 78 dB (6000 Hz) and 44 dB (700 Hz) in Ko 's study[15] and the proposed sensors were 79 dB (2000 Hz) and 50 dB (100 Hz), respectively. The average signal-to-noise ratios of Ko et al. and proposed sensor were 66 and 64 dB, respectively. However, the proposed sensor is very flat compared to Ko's sensor in the entire audible band, which is a desirable feature for a hearing aid microphone.
R1-3. The schematic in Figure 2 is very busy. Authors should provide a clearer illustration to help readers to understand the sensor structure.
Response to R1-3: Agreed. We have revised the figure for greater clarity.
R1-4. In numerical study, the mesh independence study is missing in the current manuscript.
Response to R1-4: Based on your advice, we have added the following information about the FEA modelling to the manuscript.
<Revised> P.7 Line 240
The mesh of the sensor model consisted of 23,3105 domain elements, 22,532 boundary elements and 1,992 edge elements using a defined "free tetrahedral".
R1-5. Besides of sensitivity, information about the quantification limit, reproducibility, repeatability, stability, and their correlated error bars must be included.
Response to R1-5: In this study, we proposed a hybrid sensor prototype that can improve the low frequency characteristics by replacing the mass of the existing acoustic sensor with an ECM, thereby demonstrating the feasibility of the hybrid sensor. Therefore, we believe that measuring and presenting data such as on the reproducibility, repeatability, stability, and correlated error bars of sensors by manufacturing multiple sensors is beyond the scope of this research. Once a manufacturing process is developed, a larger scale study such as this can developed with the requested data measured and reported. A brief comment is added at the end of the conclusions section.
<Revised> from P.15 Line 442
Further studies using modeling simulation and cadaveric experiments on the changes in the frequency characteristics of the sensitivity depending on the coupling strength between the sensor and the umbo are needed.
Some minor typos are shown below:
(a). In line 217, it should be “the angle θ of beam was changed from 30 degree to 60 degree”, instead of “30 degree to 5 degree”.
(b). In line 219 and 220, it should be “W=350um, T=200um, and θ=50 degree” instead of “length = 50 degree”
(c). In line 222, authors should double check the length. Is it really 50um? From Figure 6 (B), it does not look like that.
<Answer>
Agreed. We corrected all of these typographical errors.
Reviewer 2 Report
COMMENTS TO AUTHOR:
The paper presents an implementation of hybrid implantable acoustic sensor which is a practicable solution to make up the low frequency degradation of the acceleration sensor. It is a topic of interest to the researchers in the related areas but the paper needs very significant improvement before acceptance for publication. My detailed comments are as follows:
1.As is described in section 3.1, the ECM mass and the upper electrode mechanism connected to it form a second order low-pass filter (LPF). How is the order of the LPF determined?
2. The overall response of the hybrid acoustic sensor is the sum of ECM and the acceleration sensor while the noise is also superimposed at the same time. How to handle this problem?
3.Whether the installation method and the location of the hybrid sensor on the umbo which were not mentioned in the article are related to the sensor performance?
4. The paper talks about the acoustic sensor or the cochlear implant which is composed of an EMC and acceleration sensor. However, the title of the paper doesn’t emphasis the words such as acoustic or cochlear implant.
5. Your manuscript needs careful editing and pay attention to the sentence structure, subscript (such as f2H), and the tiny descriptions in the figures.
Author Response
Reviewer #2
R2-1. As is described in section 3.1, the ECM mass and the upper electrode mechanism connected to it form a second order low-pass filter (LPF). How is the order of the LPF determined?
Response to R2-1: This is a good point. Thank you. We've added the following description:
<Original> from P.4 Line 138
The vibration characteristics of the mass coupled with the vibration beam yields an LPF for the following reasons.
<Revised> from P.4 Line 155
The only difference between proposed and sensor in Fig. 1(b) is that the dummy mass is replaced with the ECM mass. The vibration characteristics of the ECM mass coupled with the vibration beam can be described a second order LPF that has each single coefficient the elastic modulus k, the mass M and the spring and the loss coefficient r.
R2-2. The overall response of the hybrid acoustic sensor is the sum of ECM and the acceleration sensor while the noise is also superimposed at the same time. How to handle this problem?
Response to R2-2: The noise floor as shown in Fig. 12 are due to the thermal noise of the FET amplifier of each capacitive sensor. The noise level of suggested hybrid sensor is expected to be twice that of a sensor using a single amplifier. In this paper, we focused on the method of compensating the frequency characteristics of each sensor to measure acoustic and vibration. As shown in Fig. 13, it can be seen that the increase of sensitivity by the hybrid sensor is more than twice as high as increase of the noise level by superimposition in the lower frequency range.
R2-3. Whether the installation method and the location of the hybrid sensor on the umbo which were not mentioned in the article are related to the sensor performance?
Response to R2-3: The hybrid sensor of the present work is based on the premise that the sensor is placed on the umbo at the posterior part of the tympanum similar to Ko et al. and shown in Fig. 1(a). When installing a sensor to the umbo, we can use a clip. But, we used glue to attach the sensor to umbo in this study. If the glue has negligible mass and very high stiffness, then both attachment methods should produce similar results. Actually, a clip has a frequency characteristics, then it is influenced to performance of sensor even though it is very small. However, it is considered that the characteristics change due to the coupling of the sensor to the umbo should be clarified through a further study based on the modeling. A brief comment is added at the end of the conclusions section.
<Revised> from P.15 Line 442
Further studies using modeling simulation and cadaveric experiments on the changes in the frequency characteristics of the sensitivity depending on the coupling strength between the sensor and the umbo are needed.
R2-4. The paper talks about the acoustic sensor or the cochlear implant which is composed of an EMC and acceleration sensor. However, the title of the paper doesn’t emphasis the words such as acoustic or cochlear implant.
Response to R2-4: Good point. Thank you. The title was modified as follows: A vibro-acoustic hybrid implantable microphone for middle ear hearing aids and cochlear implants
R2-5. Your manuscript needs careful editing and pay attention to the sentence structure, subscript (such as f2H), and the tiny descriptions in the figures.
Response to R2-5: We agree that our writing needed improvement. Our American colleagues have provided edits that we hope has improved the writing.
Reviewer 3 Report
Title: A hybrid implantable sensor with an electret condenser microphone as the mass of the acceleration sensor
Authors: Ki Woog Seong, Ha Jun Mun, Dong Ho Shin, Jong Hoon Kim, Hideko Heidi Nakajima, Sunil Puria, Jin-Ho Cho
In this paper, the author and co-authors have proposed a broadband hybrid implantable acoustic sensor for implantable hearing aid. The frequency response range of the proposed hybrid implantable acoustic sensor is wider than that of the traditional MEMS sensors. Moreover, the proposed hybrid implantable acoustic sensor owns a higher sensitivity. Meanwhile, the entire work is systematic and completed. In my view, this manuscript is suggested to be published if the following questions are answered:
1. The authors claim that the proposed broadband hybrid implantable acoustic sensor has a higher sensitivity than the traditional MEMS sensors at low frequencies. But no comparison between two kinds of sensors is given. A comparison or discussion should be provided to clarify the advantage.
2. At about 300 Hz, the frequency response of the sensor has a sudden drop. What is the reason?
3. What are the main factors affection the system noise flour?
4. The font in every figure should be enlarged.
Author Response
Reviewer #3
R3-1. The authors claim that the proposed broadband hybrid implantable acoustic sensor has a higher sensitivity than the traditional MEMS sensors at low frequencies. But no comparison between two kinds of sensors is given. A comparison or discussion should be provided to clarify the advantage.
Response to R3-1: Agreed. Conventional sensors have many problems such as body noise pick-up and surgical difficulties. Umbo attachment sensors (e.g., Ko et al.) provide solutions to these problems. In this paper, we have added a comparison Figure (13b) with Ko's study even though their results from a simulation model rather than in from biological tissue. The following text in Figure 13b was added to 3rd parapraph in P. 13.
<Revised> from P.13 Line 397, Figure 13. The output of the combined sensor output and noise floor are shown in Figure 13b. Also shown for comparison are Ko's sensitivity and noise results from a simulation model calculation (not in a cadaveric experiment). The measured noise level averaged across frequency of the proposed sensor was about 5 dB lower than that of Ko et al., and the sensitivity of Ko et al. was 1.5 dB higher. The maximum and minimum of the signal-to-noise ratios were 78 dB (6000 Hz) and 44 dB (700 Hz) in Ko 's study[15] and the proposed sensors were 79 dB (2000 Hz) and 50 dB (100 Hz), respectively. The average signal-to-noise ratios of Ko et al. and proposed sensor were 66 and 64 dB, respectively. However, the proposed sensor is very flat compared to Ko's sensor in the entire audible band, which is a desirable feature for a hearing aid microphone.
R3-2. At about 3000 Hz, the frequency response of the sensor has a sudden drop. What is the reason?
Response to R3-2: The acceleration sensor of the present work has a simple second-order mechanical model with single mass-spring damper. In the simulation, there is a single resonance as shown in Fig. 5, and the resonance frequency is the same as that of vibration beam plate. However, in the actual laboratory production, the beam plate may be slightly different from that of the design, thereby generating multiple harmonics at a specific frequency and exhibiting unexpected characteristics. The acquisition software used in this experiment collects only the magnitude at a specific frequency through FFT. Therefore, the dip as shown in Fig. 8 may occur at a certain frequency in the case of including multiple harmonics, but it is not considered to be a problem in observing the overall frequency response.
<Original> from P.8 Line 281
The FEA showed a difference of about 0.1 kHz and a low peak of the resonance characteristic. However, this is a negligible error that might be due to the manufacturing process of the vibration beam.
<Revised> from P.9 Line 311
The measurement result with fabricated vibration beam plate is similar trends to its simulation in figure 5. (b), but only difference is unexpected dip in 3 kHz. It seems to fabrication process error but it is not considered to be a big problem while observing the overall frequency response.
R3-3. What are the main factors affection the system noise flour?
Response to R3-3: The two noise sources in these experiments are acoustical noise and vibrational noise. In the cadaveric experiments, environmental noise was carefully controlled by the use of a vibration isolation table and the acquired signals were synchronously averaged 20 times to further reduce noise. The noise level as shown in Fig. 13 might be caused by this kind of environmental noise and also the thermal noise of the FET to detect the signal due to capacitance change of each sensor.
R3-4. The font in every figure should be enlarged.
Response to R3-4: Agreed. We have adjusted the font of all figures to enhance readability.
Round 2
Reviewer 1 Report
The revised manuscript has addressed most of my comments. This manuscript is well organized and written now. In summary, I recommend it be accepted.